# Rare Breast Cancers Review

**DOI:** 10.3390/healthcare12232483

**Published:** 2024-12-09

**Authors:** Bowen Song, Harnoor Singh

**Affiliations:** Department of Diagnostic and Interventional Imaging, The University of Texas Health Science Center at Houston, Houston, TX 77030, USA; bowen.song@uth.tmc.edu

**Keywords:** rare breast cancer, epithelial breast tumors, non-epithelial breast tumors, targeted therapies, molecular profiling, immunotherapy

## Abstract

Background/Objectives: Breast cancer is one of the most common malignancies in women, with rare subtypes presenting unique clinical challenges. This review provides a comprehensive analysis of rare breast cancers, including both epithelial and non-epithelial subtypes, and explores their epidemiology, pathology, prognosis, and treatment approaches. Methods: A systematic review was conducted focusing on recent advancements in the treatment of rare breast cancer subtypes. Articles were selected based on criteria emphasizing studies from the past five years, with older foundational studies included where necessary. The analysis incorporated molecular profiling, clinical trials, and advancements in targeted and immunotherapies, where possible. Results: Rare epithelial subtypes, such as tubular, mucinous, and medullary carcinomas, demonstrate distinct clinical and pathological features, with generally favorable prognoses compared to invasive ductal carcinoma (IDC). Non-epithelial cancers, including sarcomas and primary breast lymphomas, require individualized treatment due to aggressive behavior and poor prognosis in certain cases. Recent advancements in targeted therapies (e.g., HER2 inhibitors, PI3K inhibitors, and PARP inhibitors) and immunotherapies (e.g., PD-1 inhibitors) have shown promise in improving outcomes for specific molecularly characterized subtypes. Conclusions: While the management of common breast cancers has become increasingly sophisticated, rare subtypes continue to pose challenges due to limited research and small patient populations. Advances in molecular profiling and next-generation sequencing are pivotal in identifying actionable mutations and expanding personalized treatment options. Future research should focus on clinical trials and collaborative efforts to refine treatment strategies and improve outcomes for these rare subtypes.

## 1. Introduction

Breast cancer is the second most common type of cancer in women, and some studies suggest it may soon surpass all others in incidence [1,2,3]. This increase is largely attributed to advancements in diagnostic and staging technologies, which have enhanced the detection of rare subtypes of breast cancer. As a result, while the overall incidence of breast cancer continues to rise, there has been a proportional decline in tumors of unknown origin [2,4]. Among the subtypes of invasive breast cancer, invasive ductal carcinoma (IDC) remains the most prevalent, accounting for 55–80% of cases, followed by invasive lobular carcinoma (ILC), which makes up about 10% [5,6,7]. Making up the remaining 10–35% of invasive breast cancers are rarer subtypes which are broadly classified according to whether they are epithelial or non-epithelial. Rare epithelial breast cancers include tubular carcinoma, mucinous carcinoma, medullary carcinoma, papillary carcinoma, micropapillary carcinoma, and apocrine carcinoma [7,8]. Rare non-epithelial breast cancers include sarcomas, phyllodes tumors, and lymphomas of the breast. Although the management of common subtypes like IDC and ILC has advanced, treatment guidelines for rarer breast cancers are still broad and largely influenced by factors like tumor size, stage, and metastasis [3,5,9]. The goal of this review, therefore, is to present an overview of the characteristics of the aforementioned rare subtypes of invasive breast cancer and their respective treatment guidelines. Through a standardized methodology, we focused on the recent, relevant literature, ensuring that the evidence reflects current advancements. In the process, we hope to identify both areas in which we are succeeding and areas that could benefit from further research.

## 2. Methodology

### 2.1. Selection of Articles

For this review, we established clear inclusion criteria to focus on the most relevant and recent literature concerning rare breast cancers. We primarily sought to include articles published within the last five years to ensure the information reflects current understanding and advancements in the field. For articles that provide background information or technological descriptions of these rare cancers, we favored newer publications but accepted those older than five years if newer, more specific primary sources were not available. Given the rarity of these breast cancers, we did not exclude small case reports; nevertheless, we prioritized large, multicenter studies with quantitative analyses when available, as they offer more comprehensive insights into treatment efficacy and patient outcomes. This methodology allows us to present a well-rounded overview of the available evidence while addressing the unique challenges posed by the study of rare breast cancers.

### 2.2. Criteria for Description of Rare Breast Cancers

In this review, each rare breast cancer subtype is described according to the following standardized criteria to ensure comprehensive and consistent analysis across subtypes.

#### 2.2.1. Incidence and Epidemiology

Prevalence data for each subtype are provided as a percentage of all invasive breast cancers, offering context for the rarity of each type. Typical patient demographics, including age range, gender, and any unique risk factors, are detailed to highlight the populations most affected by these cancers.

#### 2.2.2. Clinical Presentation

Common presenting features, such as tumor size, location, texture, and any associated symptoms, are outlined to assist in distinguishing these subtypes from more common forms of breast cancer. Notable signs, including nipple discharge, lymph node involvement, and presence of skin changes, are described where relevant.

#### 2.2.3. Pathological and Histological Features

In analyzing the pathological and histological features of rare breast cancers, we assess cellular characteristics, growth patterns, immunohistochemical profiles, molecular markers, and histologic grading. Each subtype is defined by distinct cell structures, such as the tubules in tubular carcinoma or lipid content in lipid-rich carcinoma, alongside unique growth patterns like the sieve-like structure of cribriform carcinoma or lymphovascular invasion seen in micropapillary carcinoma. Immunohistochemistry helps identify hormone receptor status, HER2 overexpression, and subtype-specific markers (e.g., androgen receptors in apocrine carcinoma), guiding targeted therapies. Molecular profiling identifies genetic alterations like ETV6-NTRK3 fusions or PIK3CA mutations, revealing additional therapeutic targets. Histologic grading using the Nottingham system and evaluation of lymphovascular invasion provide insights into aggressiveness and potential metastatic behavior. Together, these histopathological assessments inform diagnosis, prognosis, and personalized treatment strategies for each rare subtype, underscoring the need for precise and specialized evaluation. 

#### 2.2.4. Prognosis

Survival statistics, including five-year and ten-year survival rates, are provided where available to give an overview of each subtype’s relative aggressiveness. Factors influencing prognosis, such as tumor size, lymph node involvement, hormone receptor status, and grade, are discussed to provide insight into outcomes.

#### 2.2.5. Treatment Guidelines, Controversies, and Emerging Therapies

Each subtype’s primary treatment modalities, including surgical interventions, radiation, and systemic therapies, are described based on the available evidence. Where relevant, specific techniques (e.g., breast-conserving surgery, sentinel lymph node biopsy) and their efficacy are discussed to provide an overview of standard care practices. Areas of debate, such as the need for axillary staging, the role of adjuvant chemotherapy, and the necessity of aggressive surgical margins, are explored. Differences in management approaches, particularly between cases with favorable and unfavorable prognostic factors, are examined with references to key studies. The need for more tailored treatment approaches, particularly in cases with limited evidence, is emphasized. Emerging treatment modalities, including targeted therapies and immunotherapies, are described for each subtype where applicable. The role of specific biologic agents, such as HER2-targeted therapies (e.g., trastuzumab) and PI3K inhibitors (e.g., alpelisib), is discussed in the context of subtypes that express corresponding molecular markers. Immunotherapies, such as PD-1/PD-L1 inhibitors, are highlighted in subtypes with high expression of immune checkpoint markers or other immune-responsive features. For subtypes exhibiting genetic alterations, targeted therapies like NTRK inhibitors are considered, and experimental approaches (e.g., peptide receptor radionuclide therapy for neuroendocrine tumors) are reviewed based on clinical trials. The potential benefits and limitations of these therapies are outlined, along with ongoing research to support their use and identify patients who might benefit the most. Areas where current evidence is sparse, particularly in the application of immunotherapies and biologic agents, are identified. Recommendations for future studies, including larger multicenter trials and studies focusing on molecular profiling, are emphasized to support personalized and precise treatment strategies for these rare breast cancers.

## 3. Overview of Rare Epithelial Breast Cancers: Epidemiology, Clinical Characteristics, and Treatments

### 3.1. Tubular Carcinoma

Tubular carcinoma of the breast is a very rare subtype of breast cancer that makes up approximately 1–4% of all invasive breast cancers [7,10,11,12]. It is most likely to occur in older patients ranging from 54 to 60 years of age and usually presents as a small (<1 cm), firm, or hard mass [11]. Prognosis of tubular carcinomas of the breast is generally good unless they are mixed or multifocal, with a 5-year survival of 97.1% and a 7-year survival of 95% [10,11,12,13]. Pure tubular carcinomas do not often metastasize or spread to lymph nodes but consensus on axillary exploration and sentinel lymph node biopsy is controversial. If resected, tubular carcinomas do not often reoccur. Treatment of tubular carcinoma currently involves breast-conserving surgery with adjuvant radiation therapy [14]. One study recommends axillary exploration even for small tumors < 1 cm in size [10]. Others suggest that there is no evidence that adjuvant therapy following a positive finding upon axillary exploration influences survival [15]. Adjuvant systemic therapy may not justified in routine management [12].

### 3.2. Mucinous Carcinoma

Mucinous carcinoma of the breast is a very rare subtype of breast cancer that makes up approximately 1–6% of all invasive breast cancers [16]. It is most common in older patients ranging from 55 to 60 years of age [10,16,17]. Mucinous carcinomas are variable in size, ranging anywhere from 1 to 20 cm, and may look like a gelatinous lesion with pushing margins. The average size of mucinous lesions is 3 cm [18]. Lymph node involvement is rare, and patients with mucinous carcinoma of the breast generally have a good prognosis unless there are axillary metastases or the tumor is large, with a 5-year survival of 92.3% and a 7-year survival of 91.2% [10,16,19]. Known factors that are associated with axillary metastases in patients with mucinous carcinoma of the breast include younger age, aneuploidy, high nuclear grade, and negative estrogen receptor status [16]. Patients with mucinous carcinomas are candidates for breast-conserving therapy, but recommendations regarding axillary staging are once again controversial, with one study suggesting there is survival benefit to staging tumors < 1 cm and another suggesting there is no survival benefit to staging tumors < 1 cm [10,20,21].

### 3.3. Medullary Carcinoma

Medullary carcinoma of the breast is a rare subtype of breast cancer that makes up approximately 1–6% of all invasive breast cancers [22]. It presents most commonly in middle-aged patients ranging from 45 to 55 years of age [22]. Medullary carcinomas of the breast have a lower rate of lymph node involvement compared to IDC, but there is still intermediate risk that ranges from a rate of 19–46% [23]. These tumors typically present as a unilateral, well-circumscribed, moderately firm mass that is fleshy, gray-tan, and nodular, with foci of hemorrhage, necrosis, or cystic degeneration, ranging from 2 to 3 cm in size [23,24]. In addition, they commonly present bilaterally if there is a significant family history [25]. The prognosis of medullary carcinomas is impacted by the presence of lymph node invasion and tumor size but is generally good with a 10-year survival of 74% overall and a 10-year survival of 90% if the disease does not involve the lymph nodes [26,27,28]. Treatment of medullary carcinoma is similar to that of IDC and typically involves modified or radical mastectomy with radiation and or chemotherapy [29]. For tumors that are smaller than 3 cm, breast-conserving surgery followed by radiation may be appropriate [23].

### 3.4. Papillary Carcinoma

Papillary carcinoma of the breast is a very rare subtype of breast cancer that accounts for approximately 0.5–2% of all invasive breast cancers [7,30,31]. It presents most commonly in older patients aged 55–66 years old [31,32]. Papillary carcinoma stands out from other types of invasive breast cancers in that there are currently over 100 known histologic subtypes [31]. Lymph node involvement varies depending on the histologic subtype, but the overall rate is 13.6% [32]. Patients may present with sanguineous nipple discharge or ulcerations in the case of advanced disease [5]. The prognosis of papillary carcinoma also varies depending on its histologic subtype but overall 5-year survival is 75% and 10-year survival is 60.6% [31,33]. After treatment, recurrence risk is low [31]. Current treatment guidelines recommend surgical excision for all papillary breast lesions due to the increased likelihood of missing an invasive component on core needle biopsy. Core needle biopsy was associated with a 38% rate of upgrade after surgical excision [34]. Historically, patients have received surgery only, surgery with radiation, and surgery with endocrine therapy [31]. There is no current definitive guideline for treatment of papillary carcinomas, possibly due to the need for individualized management based on histologic subtype.

### 3.5. Micropapillary Carcinoma

Micropapillary carcinoma of the breast is a very rare subtype of breast cancer that accounts for approximately 0.9–2% of all invasive breast cancers [7,35]. It presents most commonly in older patients between 50 and 60 years of age [35]. Micropapillary carcinoma is characterized and differentiated from papillary carcinoma by frequent lymphovascular invasion and nodal metastases [36]. It frequently presents as a palpable breast mass, often in a more advanced stage [35]. Prognosis is very poor, with very high incidence of nodal metastasis and very high rates of local recurrence after treatment. One study reported the rate of nodal metastasis to be as high as 66.7% and others reported a recurrence rate ranging between 55.2 and 79.6% [36,37,38,39,40]. Reported 5-year survival rates range from 72 to 92.3% and 10-year survival rates range from 55 to 84.3% [41,42,43,44]. Current treatment recommendations suggest aggressive preoperative neoadjuvant chemotherapy due to the aggressive nature of this disease process [37]. Historically, micropapillary carcinoma has been treated with modified radical or total mastectomy. More recent conservative techniques suggest regional excision with larger surgical margins and radiation therapy to prevent recurrence [45]. Interestingly, micropapillary carcinomas are also associated with higher rates for hormonal positivity relative to IDC. In particular, 35% of patients are HER2-positive, although there is no reported difference in survival in HER2-positive and HER2-negative groups [38,40,45,46,47,48,49].

### 3.6. Apocrine Carcinoma

Apocrine carcinoma of the breast is a rare subtype of breast cancer that accounts for approximately 1–4% of all invasive breast cancers [7]. It presents most commonly in older patients between 55.5 and 58.5 years of age [50,51]. Apocrine carcinoma often resembles IDC in presentation as a palpable mass that may be associated with bloody nipple discharge [52,53,54]. On histology, apocrine carcinomas may present as columnar, cuboidal, or flattened cells depending on their location [55]. Prognosis of apocrine carcinoma of the breast is uncertain and seems to vary based on the presence or absence of hormonal markers and tumor stage. Some studies have found triple-negative apocrine carcinomas to have favorable prognoses to triple-negative breast cancers with no specific type, whereas others suggest there is no significant difference [56,57,58,59,60,61]. About 50% of apocrine cancers express HER2 and are associated with a favorable prognosis [53,62]. The treatment of apocrine carcinomas is typically surgery, but there is not yet a defined guideline for the extent of surgery. Given the possibility of nodal metastasis and lymphatic invasion, sentinel node biopsy should always be considered [63]. The efficacy of neoadjuvant chemotherapy appears to depend on hormonal receptor status. For those that are HER2-positive, chemotherapy has a good response [64]. For those that are HER2-negative, chemotherapy has a limited benefit on the prognosis of the disease [65]. Finally, for those that are triple-negative, adjuvant chemotherapy is always recommended [66,67].

### 3.7. Cribriform Carcinoma

Cribriform carcinoma is a rare subtype of invasive breast cancer that represents only 0.3% percent of all breast cancer cases [68]. The slow-growing nature of the tumor allows it to have a favorable prognosis, and the demographic typically affected is postmenopausal women [69]. The treatment of cribriform carcinoma includes breast-conserving surgery, and axillary staging remains controversial, similar to tubular carcinoma [70]. The prognosis is excellent, with high survival rates since it tends to present with smaller tumor sizes, lower histologic grades, and higher rates of estrogen receptor and progesterone receptor positivity compared with the more common types of invasive breast cancer like invasive ductal carcinoma (IDC). Cribriform carcinoma has notably better survival outcomes than IDC: the disease-specific survival (DSS) and overall survival (OS) rates are significantly higher, with studies reporting a five-year DSS of 98.8% and OS rate of 95.3% [71]. Further, cribriform carcinoma has a lower frequency of axillary nodal metastases, which additionally supports its favorable prognosis [69].

### 3.8. Adenosquamous Carcinoma

Adenosquamous carcinoma of the breast is an uncommon variant of metaplastic breast cancer that features glandular and squamous components. It is an aggressive disease associated with poor prognosis and high rates of recurrence [72,73]. The primary treatment modality is commonly surgical excision, often followed by adjuvant chemotherapy. Due to the rarity of the cancer, the optimal treatment strategy is unclear, but aggressive surgical management is almost always recommended [74]. Specific molecular markers such as PD-L1 may lead to consideration of target therapies or immunotherapies. However, this evidence is still emerging and these therapies are considered on a case-by-case basis [75].

### 3.9. Neuroendocrine Carcinoma

Neuroendocrine carcinoma of the breast is a rare subtype characterized by endocrine cells making up over 50% of the tumor and by the expression of chromogranin and synaptophysin markers. Because of the high frequency of hormone receptor positivity, hormonal therapy is a potential treatment option for those patients [76,77]. However, because of its aggressive nature, most patients will need systemic chemotherapy and radiotherapy. Anthracycline and taxane-based chemotherapies are common, as are combinations of platinum compounds and etoposide, particularly for high-grade tumors [76]. For metastatic cases specifically, peptide receptor radionuclide therapy (PRRT) is currently being tested in ongoing trials, but it has not yet become a standard treatment. Tumors expressing somatostatin receptors, which are frequently expressed in neuroendocrine tumors, are targeted by this approach [78,79].

### 3.10. Lipid-Rich, Oncocytic, and Sebaceous Carcinomas

These three breast carcinomas are extremely rare and present with the following unique histopathologic features:Lipid-rich carcinoma is characterized by abundant intracellular lipids. This carcinoma has a high rate of metastasis, so it is typically associated with a poor prognosis. Management often mimics invasive ductal carcinoma with surgery and systemic therapy. In a 2015 case report from Yantai Yuhuangding Hospital, two patients with lipid-rich carcinoma underwent modified radical mastectomy and then proceeded with two distinct treatment regimens. The first patient had six cycles of chemotherapy, radiotherapy, and endocrine therapy with anastrozole. The second patient had three cycles of chemotherapy without radiotherapy. A follow-up period of 13 months and 25 months, respectively, demonstrated no evidence of recurrence or distant metastasis in either patient [80].A distinguishing feature of oncocytic carcinoma is cells that contain eosinophilic granular cytoplasm from the accumulation of mitochondria. These tumors tend to be high-grade and often overexpress HER2/neu. Treatment is usually surgical; systemic therapy is also utilized based on the tumor’s receptor status and grade [81].Sebaceous carcinoma presents with sebaceous differentiation, in which cells have vacuolated cytoplasm and peripherally displaced nuclei. Because of the rarity of this carcinoma, there are no current treatment protocols established, and management typically mirrors that of invasive ductal carcinoma, including surgery and systemic therapy [82,83]. A 2006 case report demonstrated the use of modified radical mastectomy and also highlighted the tumor’s immunohistochemical profile, which often presents with positivity for cytokeratin and epithelial membrane antigens [84].Given the rarity of these tumors, protocols of treatment are not well established; therefore, management is often similar to other common breast cancer subtypes and is dominated by surgery and systemic therapy. Although targeted therapies and immunotherapies exhibit promise in specific molecular settings, their role remains to be defined.

### 3.11. Adenoid Cystic Carcinoma

The histopathological features of adenoid cystic carcinoma of the breast are similar to adenoid cystic carcinoma of the salivary glands. Middle-aged women are most commonly affected; however, these tumors can present in any age group. The breast mass in patients is usually slow-growing, well circumscribed, and occasionally painful or tender, markedly different from other breast carcinomas. The biphasic structure is a hallmark of adenoid cystic carcinoma in which epithelial and myoepithelial cells in tubular and cribriform patterns are formed. Perineural invasion is a rare feature of this condition and would suggest a more aggressive disease. Since lymph node involvement is rare, sentinel lymph node biopsy is not routinely recommended unless suspicious findings are noted. Treatment generally consists of breast-conserving surgery or mastectomy. In cases with perineural invasion or with aggressive histologic features, radiotherapy may be necessary post-surgery. In the setting of metastasis, chemotherapy is often considered, though its role is not yet well defined due to the rarity of the disease. ACC is generally very favorable in terms of prognosis, with 10-year survival rates of 85–90%. However, local recurrence is possible, especially with perineural invasion, and although rare, distant metastases can involve the lung and other organs [85,86].

### 3.12. Secretory Breast Carcinoma

Secretory breast carcinoma is a very rare subtype of breast cancer, representing less than 0.15% of all breast cancers. It has historically been known as “juvenile carcinoma” due to its high prevalence in younger patients, but it can occur in any age group. The most common presentation is as a painless, slow-growing mass that is often confused with benign lesions [87,88]. Secretory breast carcinoma is histologically characterized by abundant secretory material seen within both intracellular and extracellular spaces. The ETV6-NTRK3 gene fusion is a defining feature of the tumor and plays a critical role in its development. The prognosis is generally favorable, with a 10-year survival rate of about 90%, especially in pediatric cases. However, adult cases seem to have a higher risk of local recurrence, so close surveillance is necessary after treatment. Furthermore, although rare, metastasis has been noted in a minority of cases [89]. Treatment consists mainly of surgery (i.e., lumpectomy or mastectomy), and sentinel lymph node biopsy is not usually performed unless clinically indicated, secondary to the low incidence of nodal involvement. Post-surgery radiotherapy can be used to reduce the risk of local recurrence. Chemotherapy is rarely indicated unless the disease shows progression or metastasis [90].

Please refer to Table 1 and Table 2 for a summary of the epidemiological and clinical characteristics and treatments of epithelial breast tumors, respectively.

## 4. Overview of Rare Non-Epithelial Breast Cancers: Epidemiology, Clinical Characteristics, and Treatments

### 4.1. Sarcomas of the Breast

Sarcomas of the breast are very rare and make up approximately 0.1–1% of breast cancers diagnosed [93,94,95]. There are various subtypes of sarcomas of the breast, and most occur in patients who are between 50.5 and 75 years of age [93,95]. The exception is in patients who have angiosarcomas, which tend to occur in patients who are younger than 40 [96]. Lymph node involvement in sarcomas of the breast is rare, and tumors typically present as a solitary, painless breast lump without nipple discharge or skin thickening [93,94,95]. Then size of these tumors varies widely, and they can grow up to 30 cm, with a mean size of 3 cm [96]. Interestingly, males made up 4% of patients with breast sarcomas, which is more than the proportion of other breast cancers where males make up about 1% [93]. Prognosis of sarcomas of the breast depends on several factors including tumor size, tumor type, age, and treatment methodology. Smaller tumors < 5 cm were associated with better prognosis. Angiosarcomas were associated with a worse prognosis [96,97]. Patients less than 50 years of age demonstrated better survival than those over 50 [93]. Treatment recommendations for sarcomas depend on the subtype of sarcoma, but they generally require surgical resection with an emphasis on attaining negative margins [93,98]. Historically, total mastectomies have been performed to provide curative therapy, but recent studies suggest that excision with wide and negative margins provides comparable results [93]. Axillary exploration is not often recommended because sarcomas typically spread hematogenously [93]. The exception to this rule may be in patients with carcinosarcomas or angiosarcomas [93,95]. Current research on radiotherapy is not strong but suggests there may be benefits in certain instances. For patients with large tumors with positive margins, radiotherapy has been shown to have marginal survival benefit [93]. Adding chemotherapy did not improve odds of survival [93].

### 4.2. Phyllodes Tumors of the Breast

Phyllodes tumors of the breast are very rare and make up approximately 0.3–1% of breast cancers diagnosed [99,100,101]. Patients with phyllodes tumors are usually between the ages of 35.6–50 years of age [99,101]. Lymph node involvement is rare [102]. Phyllodes tumors typically present similarly to fibroadenomas but may be fast-growing [102,103]. The median size of a phyllodes tumor is 4 cm, though they can grow much larger. At >10 cm in size, they are classified as giant [102,103]. These tumors are broadly categorized into three groups: benign, borderline malignant, and malignant. The majority of phyllodes tumors behave in a benign manner, and prognosis is generally good with a 10-year survival rate of 87% [101,102]. Malignant phyllodes tumors have a poor prognosis due to their risk of metastasizing to the lungs, bones, and brain [101,102]. Treatment of phyllodes tumors is surgical excision [99]. The importance of achieving wide margins depends on how the phyllodes tumor is classified. For patients with benign phyllodes tumors, a positive margin has not been shown to be related to recurrence [99,104]. For patients with borderline or malignant phyllodes tumors, re-excision or mastectomy should be completed to ensure a negative margin is achieved [99,105]. The usefulness of adjuvant radiotherapy is controversial. Some studies suggest that it is useful in patients with borderline or malignant phyllodes to decrease the risk of recurrence [106,107]. Overall, however, radiotherapy does not affect overall survival and disease-free survival [99,106,108].

### 4.3. Primary Breast Lymphoma

Primary breast lymphoma is defined by Wiseman and Liao as requiring the following:The breast as the site of presentation.Breast tissue close to lymphatic infiltration.No disseminated disease beyond the axillary nodes.No previous diagnosis of lymphoma [109].

Primary breast lymphomas are very rare and make up 0.04–1% of breast cancers diagnosed [110,111]. They primarily affect people aged 35.6–50 years old [99,101]. Lymph node involvement is relatively common and has been reported in 13–50% of cases [112,113]. Primary breast lymphoma may be difficult to differentiate from invasive ductal carcinoma and requires immunohistochemical staining for confirmation [114]. These tumors may be bilateral or unilateral [115]. Prognosis of primary breast lymphoma is uncertain with some studies reporting 5-year survival rates of up to 89% and others reporting a survival time of only 12 months [115]. Additionally, there is no standardized treatment regimen for patients with primary breast lymphoma. A review by Jennings et al. suggests that mastectomy does not offer a survival benefit but radiation therapy offers a survival benefit in patients without lymph node involvement and chemotherapy offers a survival benefit in patients with lymph node involvement [115].

Please refer to Table 3 and Table 4 for a summary of the epidemiological and clinical characteristics and treatments of epithelial breast tumors, respectively.

## 5. Targeted Therapy and Immunotherapy

Key advances in molecular profiling have ushered in the era of targeted therapies and immunotherapies that are revolutionizing the care of breast cancer, including those for more rare subtypes.

### 5.1. HER2-Targeted Therapies

HER2 (human epidermal growth factor receptor 2) overexpression is a major driver of aggressive breast cancer subtypes, including some cases of micropapillary carcinoma and apocrine carcinoma. In HER2-positive rare cancers, targeted therapies including trastuzumab (Herceptin) and pertuzumab have yielded significant survival benefits and are amongst the most common treatment options [117]. These monoclonal antibodies block the HER2 receptor, which helps to stop cancer cells from proliferating. Tyrosine kinase inhibitors such as lapatinib are also used to treat HER2-positive tumors and inhibit downstream signaling pathways, and combining these with HER2-targeted therapy has been demonstrated to improve progression-free survival (PFS) and overall survival (OS) compared to trastuzumab alone [118]. Newer agents such as tucatinib, given with trastuzumab and capecitabine, have also shown promise in adding to the successes for those with metastatic HER2-positive breast cancer [119].

For patients with aggressive or metastatic disease, HER2-targeted therapy has shown to be very beneficial. The combination of trastuzumab with pertuzumab or other chemotherapy agents has demonstrated improved outcomes in HER2-positive breast cancers [120]. Additionally, some patients may also find a benefit from ado-trastuzumab emtansine (T-DM1), which is an antibody–drug conjugate that directly delivers cytotoxic agents to cancer cells overexpressing HER2. In the 2012 EMILIA trial, patients with previously treated HER2-positive metastatic breast cancer who were randomized to T-DM1 had an improvement in progression-free survival (PFS) and overall survival (OS) versus lapatinib plus capecitabine [121].

### 5.2. PI3K Inhibitors

Mutations in the PIK3CA gene may drive tumor growth in rare hormone receptor-positive cancers like mucinous carcinoma and tubular carcinoma. PI3K inhibitors, which include alpelisib (Piqray), have become effective treatment options for these patients. Alpelisib is a PI3K pathway inhibitor that is often used in cancers with PIK3CA mutations [122]. This drug is usually used in addition to endocrine therapy (e.g., fulvestrant) in patients with hormone receptor-positive HER2-negative disease who have seen disease progression with previous endocrine therapy. The 2019 SOLAR-1 trial demonstrated significantly improved progression-free survival (PFS) within this patient population [123].

### 5.3. PARP Inhibitors

Patients with triple-negative breast cancer (TNBC) are candidates for PARP inhibitors if they have BRCA1 or BRCA2 mutations. These mutations are common in TNBC and some rare subtypes such as triple-negative apocrine carcinoma [124]. BRCA-mutated tumors are targeted by drugs like olaparib and talazoparib, which destroy cancer cells via the DNA damage repair pathway. These therapies are a promising option for patients with limited treatment options and have shown to improve progression-free survival (PFS) over standard chemotherapy in patients with advanced breast cancer and germline BRCA mutations [124,125].

### 5.4. Immune Checkpoint Inhibitors

Immune checkpoint inhibitors are a form of immunotherapy that has transformed many cancer treatments, including breast cancer. Pembrolizumab, a PD-1 inhibitor, has shown efficacy in triple-negative breast cancer (TNBC), including rare subtypes such as triple-negative apocrine carcinoma and micropapillary carcinoma. Pembrolizumab works by blocking the PD-1 pathway, allowing cancer cells to evade the immune system. Through this inhibition, these immune checkpoint inhibitors make it possible for the T cells of the body to recognize and destroy cancer cells. When given with chemotherapy, pembrolizumab shows a benefit in patients with PD-L1-positive TNBC, as seen in the KEYNOTE-355 trial, making it a treatment option for aggressive rare breast cancer subtypes. Specifically, the trial showed that pembrolizumab plus chemotherapy improved median progression-free survival (PFS) to 9.7 months compared to 5.6 months with chemotherapy alone [126]. Furthermore, trials are ongoing to examine whether a combination of these checkpoint inhibitors with other treatments, such as PARP inhibitors and targeted therapies, would further broaden the application of immunotherapy in rare or metastatic types of breast cancers [127,128].

## 6. Axillary Staging

Several controversies surround axillary staging of rare breast cancers, especially in the context of next-generation sequencing (NGS) for identifying actionable targets such as PIK3CA mutations, BRCA1/2 mutations, and PD-L1 expression. One of the main considerations includes the de-escalation of axillary surgery. Less invasive procedures, such as sentinel lymph node biopsy (SLNB), are being preferred over axillary lymph node dissection (ALND) in an attempt to reduce morbidity without compromising oncological safety. However, the optimal strategy for patients who are initially node-positive and become clinically node-negative (ycN0) after neoadjuvant chemotherapy is unclear, resulting in heterogeneous guideline recommendations [129,130,131].

Another issue is the heterogeneity in molecular profiling. NGS can identify actionable starting mutations such as PIK3CA and BRCA1/2 and biomarkers such as PD-L1, but the clinical utility of these findings to guide axillary management is not well established. For example, guidelines from the American Society of Clinical Oncology (ASCO) emphasize the value of molecular profiling but also recognize the limitations in available evidence for its use in axillary staging [123,124].

Furthermore, integrating molecular data into surgical decisions is challenging. PD-L1 expression can guide the use of immune checkpoint inhibitors in TNBC, but it is less clear in its ability to influence axillary management. Similarly, PIK3CA or BRCA mutations can guide systemic therapy choices but do not directly provide definitive information on the amount of axillary surgery that is required [123,124].

The balance between the de-escalation of surgery and maintaining oncological safety, the integration of molecular profiling into clinical practice, and the lack of standardized guidelines for incorporating NGS findings into axillary management remain a challenge in the treatment of rare breast cancers. However, continued advancements in personalized medicine and collaborative research efforts offer the potential to refine treatment protocols and improve patient outcomes.

## 7. Challenges and Future Decisions

While targeted therapies and immunotherapies have often been useful in the treatment of more common breast cancers, their use for rarer types of breast cancer is still an area of active investigation. These tumors are infrequent, and the small number of patients limits what we can learn from clinical trials. However, as molecular profiling becomes the standard, identifying actionable mutations in these rare cancers will allow more patients to gain access to these novel treatments.

In particular, the use of next-generation sequencing (NGS) to identify actionable targets, such as PIK3CA mutations, BRCA1/2 mutations, and PD-L1 expression, will likely lead to more widespread use of targeted therapies and immunotherapies [132,133]. In order to improve outcomes for patients with rare breast cancers, personalized treatment approaches based on the molecular characteristics of each tumor are essential.

## 8. Conclusions

Our understanding of rare epithelial and non-epithelial breast cancers, though still limited compared to the more common subtypes, has advanced considerably. Each rare subtype presents distinct epidemiological and clinical profiles that impact treatment strategies and prognoses. For rare epithelial tumors, such as tubular and mucinous carcinomas, the emphasis mostly remains on individualized treatment, often involving breast-conserving surgery and radiation therapy. The debate over the necessity of axillary staging and systemic therapy highlights the need for further research to refine these approaches. Medullary carcinoma, with its intermediate lymph node involvement, generally benefits from surgical and adjuvant treatments similar to those used for invasive ductal carcinoma. Papillary carcinoma’s diverse histological subtypes call for a personalized treatment strategy, while micropapillary carcinoma’s aggressive nature necessitates a more rigorous regimen of neoadjuvant chemotherapy and localized excision. Apocrine carcinoma’s variable prognosis, influenced by hormonal receptor status, further illustrates the importance of personalized treatment plans.

The era of targeted therapies and immunotherapies has begun to revolutionize the care of breast cancer, including these rare subtypes. HER2-targeted therapies, such as trastuzumab and pertuzumab, have shown significant survival benefits in HER2-positive rare cancers, including micropapillary and apocrine carcinomas. The combination of these monoclonal antibodies with tyrosine kinase inhibitors and chemotherapy agents has improved outcomes for patients with aggressive or metastatic disease. PI3K inhibitors, like alpelisib, have also become effective treatments for hormone receptor-positive cancers with PIK3CA mutations, common in some rare epithelial tumors like mucinous and tubular carcinoma. Similarly, PARP inhibitors have emerged as promising options for patients with BRCA1/2 mutations, especially in triple-negative breast cancers and rare subtypes like triple-negative apocrine carcinoma. Immune checkpoint inhibitors, such as pembrolizumab, have also demonstrated efficacy in triple-negative breast cancers, offering new treatment options for rare subtypes like triple-negative apocrine and micropapillary carcinomas. The integration of immunotherapy with chemotherapy, particularly for PD-L1-positive tumors, has improved progression-free survival in patients with aggressive disease.

In the realm of rare non-epithelial tumors, sarcomas, phyllodes tumors, and primary breast lymphomas each present unique challenges. Sarcomas, including angiosarcomas, often require extensive surgical resection with a focus on achieving negative margins, with variable benefits from additional therapies. Management of phyllodes tumors largely hinges on their classification, necessitating precise surgical interventions and, in some cases, adjuvant therapy. The treatment of primary breast lymphoma remains complex, with evidence supporting the benefit of chemotherapy in cases with lymph node involvement and radiation therapy for localized disease.

While significant strides have been made in the treatment of rare breast cancers, it is evident that gaps remain in our understanding and management strategies. The integration of molecular profiling and next-generation sequencing to identify actionable targets, such as HER2 overexpression, PIK3CA mutations, BRCA mutations, and PD-L1 expression, will allow for more personalized treatment options. Continued research and clinical trials are essential to refine treatment guidelines, enhance patient outcomes, and ensure that the management of these rare subtypes is as effective and targeted as that for more common breast cancers. As we advance, a multidisciplinary approach, incorporating the latest advancements in targeted and immunotherapies, along with individualized treatment plans, will be crucial in addressing the complexities associated with rare breast cancer subtypes.

## Figures and Tables

**Table 1 healthcare-12-02483-t001:** Epidemiological and clinical characteristics of rare epithelial breast tumors.

Epithelial Tumors	Percentage of Cases	Age	Lymph Node Involvement	Presentation and Symptoms	Prognosis
Tubular carcinoma	1–4% [7,10,11,12]	54–60 years [11]	Rare [11,12].	Usually small, <1 cm in diameter, and firm or hard [11].	Good prognosis unless mixed or multifocal. True tubular carcinomas are relatively non-aggressive and do not often recur, spread to lymph nodes, or metastasize [11,12,13]. 5-year survival—97% 7-year survival—95% [10]
Mucinous carcinoma	1–6% [16]	55–60 years [10,16,17]	Rare [16].	May present as a gelatinous lesion with pushing margins ranging from less than 1 cm to greater than 20 cm with a mean size of 3 cm [18].	Good prognosis unless patient has axillary nodal metastases or large tumor size [16,19]. Axillary nodal metastases are associated with patients with younger age, aneuploidy, high nuclear grade, and negative ER status [16]. 5-year survival—92% 7-year survival—91% [10]
Medullary carcinoma	5–7% [22]	45–54 years [24]	Lower than IDC, rate 19–46% [23].	Typically presents as a unilateral, well circumscribed, moderately firm mass that is fleshy, gray-tan, and nodular with foci of hemorrhage, necrosis, or cystic degeneration. Typically range from 2 to 3 cm in size [23,24]. Will commonly present bilaterally if there is a significant family history [25].	Prognosis is impacted primarily by presence of lymph node invasion and tumor size but is generally good. 10-year survival overall—74% 10-year survival with negative lymph nodes—90% [26,27,28]
Papillary carcinoma	0.5–2% [7,30,31]	50–66 years [31,32]	Varies depending on histologic subtype. Overall rate of axillary metastasis found to be 14% [32].	Patients may present with sanguineous nipple discharge or ulcerations in the case of advanced disease [5].	The prognosis of invasive papillary carcinoma varies depending on its individual histologic subtype with a recent gene assay showing over 100 variations of invasive papillary carcinoma. Disease recurrence risk is relatively low [31]. 5-year survival—75% 10-year survival—61% [31,33]
Micropapillary carcinoma	0.9–2% [7,35]	50–60 years [35]	High incidence of nodal metastasis with one study reporting 67% of patients showing nodal metastasis [36,40].	Frequently presents as a palpable mass, often in a more advanced stage [35].	Poor prognosis, characterized and differentiated from papillary carcinoma by frequent lymphovascular invasion and nodal metastases [36]. Additionally associated with a high rate of local recurrence with a reported rate of 55–80% [37,38,39]. 5-year survival—72–92% 10-year survival—55–84% [41,42,43,44]
Apocrine carcinoma	1–4% [7]	55.5–58.5 years [50,51]	Lymph node metastasis likelihood depends on hormonal status with triple-negative carcinomas being associated with very high rates of nodal metastasis [51,91].	Often resembles invasive ductal carcinoma in presentation as a palpable mass which is occasionally associated with bloody nipple discharge [52,53,54].	Prognosis is uncertain but seems to vary based on hormonal markers and stage. Some studies have found triple-negative apocrine carcinomas to have favorable prognoses to triple-negative breast cancers with no specific type, whereas others suggest there is no significant difference [56,57,58,59,60,61]. About 50% of apocrine cancers express HER2 and are associated with a favorable prognosis [53,62].

**Table 2 healthcare-12-02483-t002:** Treatments of rare epithelial breast tumors.

Epithelial Tumors	Treatment Guidelines
Tubular carcinoma	Breast-conserving surgery and adjuvant radiotherapy recommended [14]. Axillary exploration guidelines unclear. One study recommends axillary exploration for tumor size < 10 mm and favorable prognosis [10]. Another suggests there is no survival benefit to axillary exploration [15]. Adjuvant systemic therapy may not justified in routine management [12].
Mucinous carcinoma	Axillary exploration is controversial. One study recommends axillary exploration for tumor size < 10 mm and favorable prognosis [10]. Another suggests that axillary node staging is not beneficial in tumors that are <10 mm [20]. Adjuvant radiotherapy and hormone therapy recommended for pure mucinous carcinoma due to high expression rates of ER and PR. Adjuvant chemotherapy may not be recommended in patients with favorable risk factors [92].
Medullary carcinoma	Treatment of medullary carcinoma is similar to that of IDC and typically involves modified or radical mastectomy with radiation and or chemotherapy [29]. For tumors that are smaller than 3 cm, breast-conserving surgery followed by radiation may be appropriate [23].
Papillary carcinoma	Surgical excision is recommended for all papillary breast lesions due to the increased likelihood of missing an invasive component. Core needle biopsy was associated with a 38% rate of upgrade after surgical excision [34]. Historically, patients have received surgery only, surgery with radiation, and surgery with endocrine therapy [31]. There is no current definitive guideline for treatment of papillary carcinoma, possibly due to the need for individualized management based on histologic subtype.
Micropapillary carcinoma	Aggressive preoperative neoadjuvant chemotherapy recommended due to aggressive behavior [37]. Historically, micropapillary carcinoma has been treated with modified radical or total mastectomy. More recent conservative techniques suggest regional excision with larger surgical margins and radiation therapy to prevent recurrence [45]. Associated with higher rates for hormonal positivity relative to IDC. Of note, 35% of patients were HER2-positive, though there is no reported difference in survival in HER2-positive and HER2-negative groups [38,40,45,46,47,48,49].
Apocrine carcinoma	The treatment of apocrine carcinomas is typically surgery, but there is not yet a defined guideline for the extent of surgery. Given the possibility of nodal metastasis and lymphatic invasion, sentinel node biopsy should always be considered [63]. The efficacy of neoadjuvant chemotherapy appears to depend on hormonal receptor status. For those that are HER2-positive, chemotherapy has a good response [64]. For those that are HER2-negative, chemotherapy has a limited benefit on the prognosis of the disease [65]. Finally, for those that are triple-negative, adjuvant chemotherapy is recommended [66,67].

**Table 3 healthcare-12-02483-t003:** Epidemiological and clinical characteristics of rare non-epithelial breast tumors.

Non-Epithelial Tumors	Percentage of Cases	Age	Lymph Node Involvement	Presentation and Symptoms	Prognosis
Sarcoma	0.1–1% [93,94,95]	50.5–75 years [93,95] Angiosarcomas are the exception and occur in younger patients with a mean age < 40 [96].	Rare [93,95]	Presents as a solitary painless breast lump without axillary lymphadenopathy, nipple discharge, or skin thickening [94,96]. Can be up to 30 cm in size, with a mean of 3 cm [96]. Males made up 4% of patients with breast sarcomas, which is more than the proportion of other breast cancers, where males make up about 1% [93].	Prognosis depends on several factors including tumor size, tumor type, age, and treatment methodology. Smaller tumors < 5 cm were associated with better prognosis. Angiosarcomas were associated with a worse prognosis [96,97]. Patients less than 50 years of age demonstrated better survival than those over 50 [93]. 19 September 2024 8:59:00 a.m.
Phyllodes	0.3–1% [99,100,101]	35.6–50 years [99,101]	Rare [102]	Presents as a firm, often rapidly growing, breast mass, which is often difficult to differentiate from a fibroadenoma. The median size of a phyllodes tumor is 4 cm, though they can grow much larger. At >10 cm in size, they are classified as giant [102,103].	These tumors are broadly categorized into three groups: benign, borderline malignant, and malignant. The majority of phyllodes tumors behave in a benign manner, and prognosis is generally good [102]. Malignant phyllodes tumors have a poor prognosis due to their risk of metastasizing to the lungs, bones, and brain [101,102]. 10-year survival—87% [101]
Lymphoma	0.04–1% [110,111]	60–65 years [116]	13–50% of cases [112,113]	May present similarly to an invasive ductal carcinoma both on physical exam and on biopsy [114]. May be bilateral or unilateral [115].	Prognosis of primary breast lymphoma is uncertain with some studies reporting 5-year survival rates of up to 89% and others reporting a survival time of only 12 months [115].

**Table 4 healthcare-12-02483-t004:** Treatments of rare non-epithelial breast tumors.

Non-Epithelial Tumors	Treatment Guidelines
Sarcoma	Treatment recommendations for sarcomas depend on the subtype of sarcoma, but generally require surgical resection with an emphasis on attaining negative margins [93,98]. Historically, total mastectomies have been performed to provide curative therapy, but recent studies suggest that excision with wide and negative margins provide comparable results [93]. Axillary exploration is not often recommended because sarcomas typically spread hematogenously [93]. The exception to this rule may be in patients with carcinosarcomas or angiosarcomas [93,95]. Current research on radiotherapy is not strong but suggests there may be benefits in certain instances. For patients with large tumors with positive margins, radiotherapy has been shown to have marginal survival benefit [93].
Phyllodes	Treatment of phyllodes tumors is surgical excision [99]. The importance of achieving wide margins depends on how the phyllodes tumor is classified. For patients with benign phyllodes tumors, a positive margin has not been shown to be related to recurrence [99,104]. For patients with borderline or malignant phyllodes tumors, re-excision or mastectomy should be completed to ensure a negative margin is achieved [99,105]. The usefulness of adjuvant radiotherapy is controversial. Some studies suggest that it is useful in patients with borderline or malignant phyllodes to decrease the risk of recurrence [106,107]. Overall, however, radiotherapy does not affect overall survival and disease-free survival [99,106,108].
Lymphoma	There is no standardized treatment regimen for patients with primary breast lymphoma. A review by Jennings et al. suggests that mastectomy does not offer a survival benefit but radiation therapy offers a survival benefit in patients without lymph node involvement and chemotherapy offers a survival benefit in patients with lymph node involvement [115].

## Data Availability

Further information regarding the data reported in this review may be found in the references below. No new data was created by this study.

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
