# Peer review of "Rare Breast Cancers Review"

_healthcare, 2024, doi:10.3390/healthcare12232483_

Round 1

Reviewer 1 Report

Comments and Suggestions for Authors

The manuscript by Song and Singh presents a review of rare breast cancers with special attention to clinical characteristics and treatments available to these malignancies.

There is no unified criteria for the histological description of the tumor types chosen. Also there is no selection of histological or imaging studies as examples.

Several rare breast tumors are missing from the review, like cribriform carcinoma, adenosquamous carcinoma, neuroendocrine carcinoma, and secretory carcinoma. Other very rare tumors with uncertain prognosis and treatment options are also missing, like lipid rich, oncocytic, or sebaceous carcinomas, just to mention a few. 

Minor:

In order to maintain a uniformity of criteria, survival percentages might be presented throughout the manuscript as rounded numbers, with no decimals. Or should the authors prefer to show one decimal in all of them.

A brief histological description of apocrine carcinoma is missing.

Comments on the Quality of English Language

SI units should be separated from its magnitude number. (1-20cm, in line 76 and subsequents).

No other special comments on quality of English.

Author Response

Comment 1: There is no unified criteria for the histological description of the tumor types chosen. Also there is no selection of histological or imaging studies as examples.

Response 1: We have have described in section 2.2 Pathological and Histological Features what our plan for this comment would be, but have not made changes in the subsections yet. We would appreciate some clarification as to whether what we have described in this section is appropriate before making edits to all of the sections, and whether "selection of histological or imaging studies as examples" means you would like us to find histological examples and point out the general characteristics of each subtype cancer in those examples.

Comment 2: Several rare breast tumors are missing from the review, like cribriform carcinoma, adenosquamous carcinoma, neuroendocrine carcinoma, and secretory carcinoma. Other very rare tumors with uncertain prognosis and treatment options are also missing, like lipid rich, oncocytic, or sebaceous carcinomas, just to mention a few. 

Response 2: Please see sections 3.7-3.12 for descriptions of additional missing cancer types.

Minor comment 1: In order to maintain a uniformity of criteria, survival percentages might be presented throughout the manuscript as rounded numbers, with no decimals. Or should the authors prefer to show one decimal in all of them.

Response minor comment 1: Survival percentages should now be rounded numbers throughout the revised draft.

Minor comment 2: A brief histological description of apocrine carcinoma is missing.

Response minor comment 2: Histological description of apocrine carcinoma added. Will update more based on our criteria from Comment 1 if it seems appropriate.

English comment 1: SI units should be separated from its magnitude number. (1-20cm, in line 76 and subsequents).

Response english comment 1: SI units now separated from magnitude numbers throughout revised draft.

Reviewer 2 Report

Comments and Suggestions for Authors

The work is well organized, with sections dedicated to each breast cancer subtype and tables summarizing symptoms, prognosis, and recommended treatments. The authors stress the need for more developments in the knowledge and treatment of rare breast tumors while acknowledging research gaps. I have a few comments regarding the publication of the manuscript.

1. The text in the introduction and abstract sections are the same. I would advise the writers to revise the abstract to make it more presentable.

2. There are additional cancer types that fall under the category of rare epithelial breast cancers, such as neuroendocrine tumors, adenoid cystic carcinoma, secretory breast carcinoma, metaplastic carcinoma, etc. Treatment strategies exist for some of these cancers. Is there a reason why the authors left out these subtypes in the current review?

3. Minor language editing is required.

      • Keep a space between the units and the numerals.

Comments on the Quality of English Language

Minor language editing is required.

      • Keep a space between the units and the numerals.

Author Response

Comment 1: The text in the introduction and abstract sections are the same. I would advise the writers to revise the abstract to make it more presentable.

Response 1: Please see revised draft, new abstract is at the top of the revised draft.

Comment 2: There are additional cancer types that fall under the category of rare epithelial breast cancers, such as neuroendocrine tumors, adenoid cystic carcinoma, secretory breast carcinoma, metaplastic carcinoma, etc. Treatment strategies exist for some of these cancers. Is there a reason why the authors left out these subtypes in the current review?

Response 2: Please see revised draft sections 3.7-3.12 for descriptions of the additional rare epithelial cancer types.

Comment 3: Keep a space between the units and the numerals.

Response 3: There should now be a space between all of the units and numerals.

Thank you very much for all of your suggestions, we really appreciate it!

Reviewer 3 Report

Comments and Suggestions for Authors

After reviewing the paper I suggest to recommend a rejection. While the topic itself is important, there are several areas where the paper falls short of expectations and could use significant improvements.

1.  The paper doesn't really bring anything new to the table. It mainly repeats what’s already widely known about rare breast cancers. Reviews should offer fresh perspectives or at least present recent developments, but this one mostly reiterates existing knowledge without identifying any real gaps in the field or suggesting new avenues for research. For example, there’s no mention of recent advancements in targeted therapies or immunotherapies, which would’ve made the review more impactful.

2. A lot of the references used are a bit old and don’t reflect the latest research. For instance, in section 2.3 on medullary carcinoma, you could’ve cited newer studies that explore updated treatment protocols. A stronger paper would pull in more recent sources and reflect the state of knowledge as it stands today, not several years ago.

3. The analysis of each subtype of breast cancer feels very surface level. While you list some statistics and general facts, the paper lacks depth. For example, in section 2.2 on mucinous carcinoma, the prognosis is only briefly mentioned, and the nuances about how factors like age or hormone receptor status impact outcomes aren't fully explored. The treatment discussion is also a bit thin — you could have gone deeper into the controversies surrounding axillary staging or emerging treatment strategies.

4. There's a lot of repetition across the sections, especially when describing the symptoms and treatment options. For example, in sections on tubular and mucinous carcinoma, the same information about surgical options and adjuvant therapy is repeated. This could’ve been condensed or streamlined to avoid redundancy and improve the flow of the paper.

5. The formatting is inconsistent in some places. For instance, the table on page 12 listing epidemiological characteristics doesn’t really add anything new and could’ve been integrated into the text instead. Also, some headings are misplaced, and paragraphs are uneven in length, which makes the paper harder to read. Some sentences are hard to follow or seem unnecessarily complicated. For example, in the introduction, the phrase "the increase in incidence of breast cancer is associated with a proportional decrease in undetermined tumors" could be clearer. I recommend simplifying the language to make the paper more readable. There’s also some overuse of technical words that might be hard for readers who aren’t experts in the field.

Overall, the paper needs to be much more detailed and updated to make a real contribution.

Comments on the Quality of English Language

Needs improvement.

Author Response

Comment 1: The paper doesn't really bring anything new to the table. It mainly repeats what’s already widely known about rare breast cancers. Reviews should offer fresh perspectives or at least present recent developments, but this one mostly reiterates existing knowledge without identifying any real gaps in the field or suggesting new avenues for research. For example, there’s no mention of recent advancements in targeted therapies or immunotherapies, which would’ve made the review more impactful. Response 1: Please see section 5: Targeted Therapy and Immunotherapy and section 6: Conclusion for discussion of the advancements of targeted therapies and immunotherapies in treating rare cancers. In the interest of not being repetitive across subsections, we tried to summarize the main treatments in this section. Comment 2: A lot of the references used are a bit old and don’t reflect the latest research. For instance, in section 2.3 on medullary carcinoma, you could’ve cited newer studies that explore updated treatment protocols. A stronger paper would pull in more recent sources and reflect the state of knowledge as it stands today, not several years ago. Response 2: In our newly written section 5, we attempt to address some of the newer protocols for treating these cancers. Would it be preferable to include these newer protocols in this section alone, or include descriptions of these protocols within each subsection?  Comment 3: The analysis of each subtype of breast cancer feels very surface level. While you list some statistics and general facts, the paper lacks depth. For example, in section 2.2 on mucinous carcinoma, the prognosis is only briefly mentioned, and the nuances about how factors like age or hormone receptor status impact outcomes aren't fully explored. The treatment discussion is also a bit thin — you could have gone deeper into the controversies surrounding axillary staging or emerging treatment strategies. Response 3: Regarding axillary staging and treatment strategies, please see section 5 and section 6. Upon looking into the controversies of staging and treatment strategies, there seemed to be a bit of overlap between subtypes of cancers, so we put them into a separate section. As in response 2, would it be preferable to include these protocols in each specific subtype or just an overall summary in these sections. Comment 4. There's a lot of repetition across the sections, especially when describing the symptoms and treatment options. For example, in sections on tubular and mucinous carcinoma, the same information about surgical options and adjuvant therapy is repeated. This could’ve been condensed or streamlined to avoid redundancy and improve the flow of the paper. Response 4: We were hoping you could provide some clarification regarding this comment. Should we be putting the treatment options and symptoms in a separate section like we have done for the above? Or would it be better to just list both tubular and mucinous carcinomas in the same paragraph and retain the existing subsections layout?
Comment 5: The formatting is inconsistent in some places. For instance, the table on page 12 listing epidemiological characteristics doesn’t really add anything new and could’ve been integrated into the text instead. Also, some headings are misplaced, and paragraphs are uneven in length, which makes the paper harder to read. Some sentences are hard to follow or seem unnecessarily complicated. For example, in the introduction, the phrase "the increase in incidence of breast cancer is associated with a proportional decrease in undetermined tumors" could be clearer. I recommend simplifying the language to make the paper more readable. There’s also some overuse of technical words that might be hard for readers who aren’t experts in the field. Response 5: In the same vein as the above response, would it be preferable to present the information just in paragraph format in each cancer's subsection, or should we present it just once in the table? Tables have not been updated with the new subtypes, but we can depending on whether you think the tables are helpful. With respect to headings, from our end they look okay. Will submit a PDF to see if that helps fix the formatting issues. Regarding your point about uneven paragraph length, could you please point us to the paragraphs that are difficult to read? Rephrased the introduction sentence.